# Arabidopsis Class II Formins AtFH13 and AtFH14 Can Form Heterodimers but Exhibit Distinct Patterns of Cellular Localization

**DOI:** 10.3390/ijms21010348

**Published:** 2020-01-05

**Authors:** Eva Kollárová, Anežka Baquero Forero, Lenka Stillerová, Sylva Přerostová, Fatima Cvrčková

**Affiliations:** Department of Experimental Plant Biology, Faculty of Sciences, Charles University, Viničná 5, CZ 128 44 Prague, Czech Republic; eva.slikova@natur.cuni.cz (E.K.); anezka.houskova@natur.cuni.cz (A.B.F.); Lstillerova@seznam.cz (L.S.); prerostova@ueb.cas.cz (S.P.)

**Keywords:** AtFH13, AtFH14, At5g58160, At1g31810, FH2 domain, class II formin, confocal laser scanning microscopy, PTEN-like domain

## Abstract

Formins are evolutionarily conserved multi-domain proteins participating in the control of both actin and microtubule dynamics. Angiosperm formins form two evolutionarily distinct families, Class I and Class II, with class-specific domain layouts. The model plant *Arabidopsis thaliana* has 21 formin-encoding loci, including 10 Class II members. In this study, we analyze the subcellular localization of two *A. thaliana* Class II formins exhibiting typical domain organization, the so far uncharacterized formin AtFH13 (At5g58160) and its distant homolog AtFH14 (At1g31810), previously reported to bind microtubules. Fluorescent protein-tagged full length formins and their individual domains were transiently expressed in *Nicotiana benthamiana* leaves under the control of a constitutive promoter and their subcellular localization (including co-localization with cytoskeletal structures and the endoplasmic reticulum) was examined using confocal microscopy. While the two formins exhibit distinct and only partially overlapping localization patterns, they both associate with microtubules via the conserved formin homology 2 (FH2) domain and with the periphery of the endoplasmic reticulum, at least in part via the N-terminal PTEN (Phosphatase and Tensin)-like domain. Surprisingly, FH2 domains of AtFH13 and AtFH14 can form heterodimers in the yeast two-hybrid assay—a first case of potentially biologically relevant formin heterodimerization mediated solely by the FH2 domain.

## 1. Introduction

The eukaryotic cytoskeleton represents a dynamic network of protein filaments and tubules that has been extensively studied in a variety of model systems. In addition to basic cellular functions, including nuclear division, cytokinesis, organelle positioning, membrane trafficking and cell expansion, the cytoskeleton plays an important role in processes such as polar cell growth, cell division plane positioning, or cell to cell communication, which are essential for proper morphogenesis and thus the development of multicellular bodies in both metazoans [1] and plants [2]. This dynamic network relies on a large ensemble of cytoskeleton-associated proteins controlling organization, remodeling, and crosstalk of cytoskeletal systems, as well as coordination of the cytoskeleton with cell membranes and organelles. While the cellular structures and organismal body plans vary greatly among the eukaryotes, many molecular and cellular mechanisms involving the two basic cytoskeletal systems—the actin microfilaments and microtubules—are conserved among divergent groups such as, e.g., plants and metazoans, which are connected only through the last eukaryotic common ancestor [3,4].

Formins, or FH2 proteins, members of an evolutionarily ancient and widely expressed family of eukaryotic cytoskeletal organizers, are a prominent example of such conserved components of the cell’s morphogenetic machinery. They are generally characterized by the presence of the conserved formin homology 2 (FH2) domain whose dimer can nucleate and cap actin filaments. In addition to other variable regulatory domains, the FH2 domain is usually accompanied by the proline-rich formin homology 1 (FH1) domain that can bind actin-profilin complexes, providing a substrate for formin-driven assembly of unbranched actin filaments [5]. However, formins also associate with microtubules via the FH2 domain or via other specialized domains, affecting microtubule dynamics [6,7,8,9]. Thus, the physiological functions of formins related to morphogenesis, cell division, cytokinesis, cell polarity, and cell to cell trafficking are likely to be tightly coupled with their roles in cytoskeleton reorganization. All eukaryotic genomes examined, including fungi, metazoans, and plants, encode numerous formin paralogs [10,11,12], most of them experimentally hitherto uncharacterized. Formins are well established to form FH2 domain dimers [13], suggesting that heterodimerization among formin paralogs might further contribute to their functional diversification. However, there is so far very little evidence of formin heterodimerization, with the only documented cases involving closely related metazoan formins [14,15,16].

The *Arabidopsis thaliana* genome harbors 21 formin-encoding loci that can be divided into two phylogenetically distinct classes present in all angiosperms, referred to as Class I and Class II, based on their sequence similarity and domain organization [11,17,18]. Most Class I formins contain a N-terminal transmembrane domain that enables them to anchor cytoskeletal structures to membranes, while many Class II formins contain a N-terminal PTEN (Phosphatase and Tensin)-like domain (homologous to members of the widespread phosphatase and tensin homolog protein family) instead. The PTEN-like domain was predicted to associate with membranes [18]. This has later been confirmed in the case of the *Physcomitrella patens* For2A formin whose PTEN-like domain interacts with PI(3,5)P_2_; this interaction is necessary for formin localization to the tip of apically growing cells [19]. Similarly, the PTEN-like domain of the rice Class II formin FH5 encoded by rice morphology determinant (RMD) gene determines tip localization of this protein in pollen tubes [20] and interestingly also mediates its anchorage to the outer surface of chloroplasts [21]. In Arabidopsis, the only typical Class II formin containing a PTEN-like domain experimentally characterized so far is AtFH14 (At1g31810), whose dimer can bind to actin barbed ends [22] and which associates with microtubules in mitotic BY-2 cells, participating in the control of mitosis and cytokinesis [23].

In this study, we compare the intracellular localization of AtFH14 with that of a hitherto uncharacterized *A. thaliana* Class II formin, AtFH13 (At5g58160) that shares overall domain structure composition with AtFH14, using *in planta* transient heterologous expression in native Australian tobacco (*Nicotiana benthamiana*) leaves. While both formins can associate with microtubules (MTs) and the endoplasmic reticulum (ER), they exhibit distinct subcellular localization patterns. In addition, we demonstrate that although these formins show different pattern of localization, their FH2 domains are capable of heterodimerization—a first such observation in plant formins.

## 2. Results

### 2.1. Construction of Fluorescent Protein-Tagged AtFH13 and AtFH14 Derivatives

Both AtFH13 (At5g58160) and AtFH14 (At1g31810) are typical Class II formins [18], containing a N-terminal PTEN-related domain, followed by a C2 domain (related to the Ca^2+^-binding domain from protein kinase C), the proline-rich FH1 domain and the C-terminal FH2 domain. Although they share the overall domain structure, AtFH13 and AtFH14 are only 52% identical in their PTEN-like and C2 domains and 60% identical in their FH2 domains, and represent branches of the Class II formin clade that are clearly separate at least within Brassicaceae (Figure 1a), and possibly within angiosperms [11]. As usual for Class II formins, both proteins are encoded by genes of a complex exon–intron structure (Figure 1b).

To examine in vivo subcellular localization of these formins, we constructed vectors expressing full length AtFH13 (Uniprot Acc. No. Q9LVN1) or a previously characterized variant of AtFH14 [23] tagged by C-terminal yellow fluorescent protein (YFP) or red fluorescent protein (RFP) fusion under the control of the constitutive ubiquitin 10 promoter (UBQ10) and used these constructs for transient transformation of *N. benthamiana* leaves. To examine contribution of individual domains of both proteins to their localization, we also prepared constructs expressing YFP-tagged N-terminal fragments of either protein containing the PTEN-like and C2 domains, as well as green fluorescent protein (GFP) and YFP-tagged isolated FH2 domains of AtFH13 and AtFH14. In addition, we also generated an YFP fusion of the N-terminal (PTEN-like and C2 domain-containing) fragment from a fortuitously cloned AtFH13 variant missing 21 amino acids within the PTEN-like domain (denoted PTENΔ) as a putative inactive variant of the PTEN domain (Figure 1c).

### 2.2. Both AtFH13 and AtFH14 Associate with Microtubules and the ER in Tobacco Epidermis

As usual in transient *N. benthamiana* leaf epidermis transformation, individual transformed cells exhibited variable signal intensity. Confocal imaging of YFP-tagged AtFH13 in cells with a relatively weak signal showed fibers, suggesting cytoskeletal association, as well as punctate structures of varying size in the cortical cytoplasm (Figure 2a), whereas cells which overexpressed the construct exhibited large, brightly fluorescent aggregates (Figure 2b). In case of AtFH14-YFP the cortical signal was mainly of fibrous character (Figure 2c), suggesting association with microtubules, consistent with previously observed localization of this protein’s FH1-FH2 domain construct to the preprophase band, mitotic spindle, and phragmoplast in tobacco BY2 cells [23].

To establish the relationship between the formin-labeled fibers and cytoskeletal structures, we co-expressed both tagged formins with the actin and microtubule markers LifeAct-RFP or KMD (kinesin motor domain)-RFP, respectively. Neither AtFH13 nor AtFH14 showed association with actin fibers (Figure 3) but both proteins differed in their localization patterns. AtFH13 exhibited patchy distribution following the microtubules, whereas AtFH14 decorated the microtubule network more homogeneously (Figure 4, Appendix A). An obvious, though statistically non-significant, trend towards co-localization with microtubules rather than microfilaments was detected also by quantitative image analysis (Appendix A).

To confirm functional connection between the AtFH13-labelled foci and the microtubule cytoskeleton, we examined the effect of the microtubule depolymerizing drug oryzalin on localization of AtFH13-YFP co-expressed with KMD-RFP. The oryzalin treatment caused partial redistribution of the cytoplasmic dots and increased their mobility, which may be due to cytoplasmic streaming, not affected by microtubule disruption (Figure 5). This suggests that integrity of microtubules is essential to maintain subcellular distribution of AtFH13.

Since both AtFH13 and AtFH14 contain the PTEN-like domain known to associate with membranes [20,24], we hypothesized that the fluorescent cytoplasmic punctate structures not associated with the cytoskeleton may be related to compartments of the endomembrane system. We thus examined co-localization of AtFH13-YFP and AtFH14-YFP with the ER marker ER-rk (ER red kanamycin). Remarkably, the two proteins exhibited obviously different localization patterns, with AtFH13-labelled structures peripherally associated with the ER network with minimal co-localization, while AtFH14-labeled structures exhibited noticeable overlap with the ER, which, however, could have been due to microtubule-ER co-alignment (Figure 6, Appendix A). Better co-localization of AtFH14 with the ER was supported also by a statistically non-significant trend detected quantitatively (Appendix A).

### 2.3. Isolated FH2 Domains of AtFH13 and AtFH14 Exhibit Distinct Patterns of Microtubule Association

Next, we compared the localization of isolated FH2 domains of AtFH13 and AtFH14 in transiently transformed epidermal cells. GFP-tagged FH2 domain of AtFH13 was predominantly cytoplasmic with a few small punctate structures; in contrast, YFP-tagged FH2 domain of AtFH14 exhibited prominent fibrous distribution similar to the full-length protein (Figure 7a). Interestingly, upon co-expression with LifeAct-RFP, the GFP-tagged FH2 domain of AtFH13 organized into fibrous structures partially co-localizing with the actin marker, whereas the YFP-tagged FH2 domain of AtFH14 retained fibrous organization clearly distinct from the actin meshwork, similar to the full-length protein (Figure 7b). The microtubular nature of the network decorated by the YFP-tagged FH2 domain of AtFH14 was also confirmed by co-expression with the microtubule marker KMD-RFP. Remarkably, also the rare cytoplasmic dots labeled by GFP-tagged FH2 domain of AtFH13 were microtubule-associated (Figure 7c, Appendix A).

### 2.4. The PTEN-Like Domain Mediates Association of AtFH13 and AtFH14 to the ER

To assess the possible contribution of the membrane-binding PTEN-like domain to the observed peripheral association of AtFH13 and AtFH14 with the ER, we examined the subcellular localization of YFP-tagged PTEN-like domains of AtFH13 and AtFH14. For control, we also included a fortuitously cloned deletion derivative of the PTEN-like domain from AtFH13, PTENΔ_FH13, presumed to be inactive due to disruption of a structurally important segment of the protein. The YFP-tagged PTEN domain of AtFH13 accumulated in structures reminiscent of the ER in the cytoplasm, whereas an analogous derivative of AtFH14 decorated small cytoplasmic punctate structures while exhibiting also strong accumulation in the nucleus and weak cytoplasmic background. In contrast, the deletion derivative of the AtFH13 PTEN-like domain showed only diffuse cytoplasmic distribution, consistent with the deletion altering the protein´s folding and consequently its subcellular localization (Figure 8a). Co-expression with the ER marker ER-rk shows that the YFP-tagged PTEN-like domain of AtFH13 peripherally associates with the ER, while its deletion derivative is cytoplasmic. The particles decorated by YFP-tagged PTEN-like domain of AtFH14 associated, and even partly overlapped, with the ER (Figure 8b).

### 2.5. FH2 Domains of AtFH14 and AtFH13 Can Heterodimerize

It is well established that formins can form homodimers through their FH2 domains [13] but there is so far only limited evidence regarding their possible heterodimerization (see Introduction). The distinct localization patterns of AtFH13 and AtFH14 suggested that if heterodimerization between these formins does take place, it is unlikely to affect a substantial part of the cellular pool of both proteins. We nevertheless tested whether FH2 domains of AtFH13 and AtFH14 can heterodimerize by means of the yeast two hybrid assay.

In the yeast two hybrid system, an isolated FH2 domain of AtFH13 can form homodimers as expected. However, in our two-hybrid setup the FH2 domain of AtFH13 also readily formed heterodimers with the FH2 domain of AtFH14 (Figure 9). This is, at the first glance, surprising, given the distinct cellular localization of these two proteins. Nevertheless, when co-expressed, full length YFP-tagged AtFH13 and RFP-tagged AtFH14 exhibited partial co-localization in tobacco epidermal cells, in spite of their different overall localization patterns (Figure 10a). In particular, the puncta of AtFH13 often co-localized with fibers decorated by AtFH14, as documented by pixel intensity analyzes of the florescence signal (Figure 10b), consistent with partial co-localization which may involve FH2 domain-mediated heterodimerization. Quantitative estimate of co-localization of the two formins yielded values significantly higher than those obtained for either formin with the non-colocalizing actin marker LifeAct and also higher than those observed for microtubules and the ER, although in these cases the differences mostly were not statistically significant (Appendix A).

## 3. Discussion

Formins are an evolutionarily ancient family of proteins [10,11,12] that can nucleate, cap, and bundle actin filaments, but also bind microtubules, resulting in an ability to modulate actin and microtubule organization and dynamics [7,9], well documented also in plants [8,25,26,27]. Angiosperm plants possess two clades of formins whose members often exhibit characteristic domain organization. While Class I formins are usually, though not always, integral membrane proteins [11,28], their Class II counterparts often harbor an N-terminal domain related to the mammalian PTEN (phosphatase and tensin homolog) lipid/protein phosphatase that was proposed to mediate membrane binding [11] and later shown to be responsible for membrane association and specific intracellular localization of Class II formins in the moss *Physcomitrella patens* [19,24] and in rice [21]. Thus, typical formins of both Class I and Class II are capable of membrane localization (reviewed in References [25,29]). In case of Class I formins, plasmalemma or endomembrane localization, as well as cytoskeletal, especially microtubule, association is well documented (e.g., References [6,30,31,32,33,34,35]). Much less is, however, known about in vivo intracellular localization of Class II formins.

In this study, we use a heterologous *in planta* expression system to document in vivo subcellular localization of fluorescent protein-tagged derivatives of two Arabidopsis Class II formins, the hitherto uncharacterized AtFH13 and somewhat characterized AtFH14 (see below). Although protein tagging by fusion with a relatively large polypeptide such as the YFP or GFP might influence its intracellular localization, such fusion proteins, including a plant formin [35], have been documented to be functional. Unfortunately, we could not confirm our observations by an independent method since we were unable to obtain specific antibodies suitable for immunostaining of formins, including AtFH13 [36]. Another theoretical alternative, expression of small tags visualized using cell permeable ligands, is well established in mammalian cell biology but practically restricted to the study of extracytoplasmic proteins in plants [37], with the exception of a single report with no published experimental follow-up [38].

Both formins studied exhibit the typical domain composition involving a N-terminal PTEN-like domain followed by a repetitive, proline-rich FH1 domain believed to mediate profilin-actin binding and by the C-terminal conserved dimerizing and actin-nucleating FH2 domain [11,18]. Despite the shared overall domain organization, the two proteins only exhibit limited similarity. Phylogenetic analysis of combined PTEN-like and FH2 domain sequences documented that they represent two distinct branches of the Class II formin clade that have separated no later than in basal Brassicaceae, and possibly earlier. While inner topology of the Class II formin clade in previous single-domain phylogenetic analyzes [11,18] had poor statistical support, it was nevertheless consistent with possible divergence of the clades represented by AtFH13 and AtFH14 as early as in ancestral angiosperms. The two proteins thus may exhibit different behavior and function, including distinct patterns of intracellular localization, which, indeed, turns out to be the case.

Formins, as known actin nucleators, can be expected to associate with actin filaments. Indeed, although AtFH14 was shown to preferentially bind microtubules, some co-localization of AtFH14 and actin was reported upon heterologous expression in mitotic tobacco BY2 cells [23]. In addition, using single molecule total internal reflection fluorescence microscopy, the FH1 and FH2 domains of AtFH14 were documented to directly bind the microfilament barbed end and act as a processive formin in vitro [22]. Nevertheless, in our *in planta* system co-expression of YFP-tagged full length AtFH14 or AtFH13 with actin markers did not reveal co-localization with the actin cytoskeleton, consistent with the strength, or indeed presence, of formin-microfilament association depending on the cellular context and molecular environment. Similar behavior has also been observed for the Class I formin AtFH4 which can bind and nucleate actin in vitro but does not decorate actin filaments in vivo [6]. Surprisingly, while the isolated FH2 domain of AtFH14 behaved comparably to the full length protein in co-expression with an actin marker, i.e., did not co-localize with actin fibers, the FH2 domain of AtFH13 showed a limited overlap with the actin network, suggesting differences in microfilament affinity between the two Class II formins.

Several Class II plant formins including AtFH14 were shown to interact with microtubules, sometimes more readily than with actin filaments [21,23,39,40]. We confirmed that YFP-tagged AtFH14 can co-localize with microtubules *in vivo*, and documented microtubule co-localization also for AtFH13. Remarkably, AtFH13 decorated the microtubules in a noticeably more discontinuous manner than AtFH14, often forming distinct particles along the microtubules. Mobility of these particles was greatly increased by the MT-depolymerizing drug oryzalin, proving that their organization is microtubule-dependent. Moreover, while the isolated FH2 domain of AtFH14 readily localized to the microtubules, the FH2 domain of AtFH13 was predominantly cytoplasmic, clearly indicating functional differences between the two FH2 domains. Analogously, metazoan formins have been documented to vary in their abilities to interact with microtubules and showed distinct localization patterns [41,42].

Interestingly, while both AtFH13 and AtFH14 associated with microtubules, only in the later the FH2 domain, which is also responsible for actin binding, was sufficient for microtubule interaction. Direct binding of FH2 domains to microtubules, or at least a specific contribution of FH2 to microtubule interaction, has been reported for several metazoan formins [41,43,44,45], as well as for plant formins of both Class I, such as rice FH15 [46], and Class II, namely rice FH5 [39]. In other cases, notably including Arabidopsis Class I formins AtFH4 and AtFH8 [6], the formin-microtubule interaction is mediated by specific domains or motifs outside FH2. In some cases, the association between formins and microtubules may be indirect, i.e., mediated by microtubule-binding proteins [47]. Together with previous reports, the observed different localization of isolated FH2 domains derived from AtFH13 and AtFH14 documents that while microtubule association appears to be a general feature of formins, its mechanism can vary substantially even within a relatively narrow clade such as the plant Class II formins.

In addition to microtubule co-localization, we observed partial overlap of AtFH14-labeled structures with the endoplasmic reticulum, as well as peripheral association of AtFH13-labelled particles with the ER network. Some derivatives of the Arabidopsis Class I formins, AtFH4 [6] and AtFH5 [25], as well as the non-conventional Class II formin AtFH16, which lacks the PTEN domain, can localize to the ER or to ER-like intracellular structures [25], and the transmembrane Class I formin AtFH1 was recently documented to pass through several endomembrane system compartments in differentiating root tissues [35]. The metazoan INF2 formin also associates with the ER periphery [48]. Our observations, which are consistent with the PTEN-like domains of AtFH13 and AtFH14 mediating peripheral association with the ER, are thus not surprising. However, the membrane nature and compartment identity of the observed formin-positive intracytoplasmic structures remains to be established.

Crystallographic studies revealed that the functional form of formin’s FH2 domain is an antiparallel dimer capable of actin nucleation or barbed end capping [13]. In the case of AtFH14, dimerization and binding of the dimer to the barbed end of microfilaments has been documented for the FH1-FH2 domain combination, as well as for the isolated FH2 domain [22]. Using the yeast two hybrid assay, we confirmed that also the FH2 domain of AtFH13 can form homodimers. Although the localization patterns of AtFH13 and AtFH14 or their FH2 domains in transiently transformed *N. benthamiana* epidermis differed noticeably, we nevertheless observed co-localization of a fraction of the signal, and documented that the FH2 domains of these two Class II formins can form heterodimers in the yeast two-hybrid assay. 

Heterodimerization of paralogous formins might theoretically provide a possible mechanism of formin activity regulation, as well as means of increasing functional diversity of fomins [18]; however, heterodimer formation may be, and often probably is, limited by structural divergence among formin paralogs [49]. Very few experimental studies address this topic, and so far, the only evidence of FH2 domain-mediated formin heterodimerization comes from metazoa. The FH2 domains of mammalian FRL2 and FRL3 are namely able to form hetero-oligomers when combined with N-terminal dimerization domains [16]. Other reported cases of mammalian Diaphanous-like formin heterodimerization [14,15] involve interactions mediated by domains other than FH2. Our finding that isolated FH2 domains of AtFH13 and AtFH14 can form heterodimers thus presents, to our knowledge, the first documented case of exclusively FH2-mediated formin heterodimerization that might further increase the functional diversity of/not only plant formins.

## 4. Materials and Methods

For phylogenetic analysis, closest AtFH13 and AtFH14 relatives were identified by BLAST in the RefSeq section of GenBank after taxonomic restriction of the database to *Arabidopsis lyrata* or to *Brassica* sp. Only full-length sequences that retrieved the original query upon reverse BLAST search as the best match were considered. Missing exons in the predicted sequence of the AtFH13 ortholog from *A. lyrata* were added manually based on an experimentally sequenced partial cDNA (GenBank EFH40810). Multiple sequence alignment was constructed, manually cleaned of unreliably aligned segments and used to compute a maximum likelihood phylogenetic tree as described previously [50] except that Gamma distribution was used to model evolutionary rate differences among sites.

Constructs for transient expression *in planta* were prepared as follows. Vectors were generated using conventional restriction cloning or the Gateway (GW) cloning system (Invitrogen^TM^, Carlsbad, CA, USA). For construction of entry clones, desired inserts were amplified using a PCR reaction with specific primer sets and templates summarized in Appendix A. To amplify the AtFH14, the previously published cDNA clone [23] has been used as a template. Enzymatic cloning was used to create all entry clones derived from the GW-compatible donor vector pENTRA^TM^1A (Invitrogen^TM^) for further construction of expression vectors. Amplified inserts were digested with appropriate restriction enzymes (Fermentas, Vilnius, Lithuania; Appendix A) and further ligated into the corresponding sites of the donor vector using T4 DNA ligase (Fermentas). For PCR-based entry clone construction, the pDONR™221 (Invitrogen^TM^) vector was used. The PCR products were amplified by specific primers (Appendix A) and transferred into donor vector by BP Clonase II (Invitrogen^TM^). To obtain expression vectors, the resulting entry clones and binary destination vector pUBC-YFP-DEST/-RFP-DEST/-GFP-DES [51], which allows C-terminal fusion of the desired protein with YFP/RFP/GFP tag were recombined using LR Clonase II (Invitrogen^TM^). The combinations of entry clones and destination vectors are summarized in Appendix A. All newly cloned inserts where verified by sequencing. 

For Agrobacterium-mediated transient transformation, expression vectors were transferred by electroporation into *Agrobacterium tumefaciens* strain GV3101 or C58C1. Plasmids carrying fluorescent protein markers, namely LifeAct-RFP [52], KMD-RFP [6], ER-rk [53], were used in co-localization experiments. Leaves from 2–3 weeks old *Nicotiana benthamiana* (native Australian tobacco) plants grown on peat pellets (Jiffy, Zwijndrecht, Netherlands) under the 8 h dark/16 h light photoperiod at 22 °C were used. Overnight *Agrobacterium* cultures were diluted in the infiltration medium (50 mM MES pH 5,6; 2 mM Na_3_PO_4_; 20 mM MgSO_4_; 0.5% glucose, 100 μM acetosyringone). A mixture containing equal amounts of bacterial suspensions carrying each construct of interest, corresponding to final optical density (OD_600_) 0.07 each, together with an amount of a suspension of agrobacteria expressing the viral silencing suppressor p19 [54] corresponding to OD_600_ 0.05, was infiltrated into the abaxial side of *N. benthamiana* leaves using a syringe. The fluorescent proteins were observed 48 h post-infiltration using confocal microscopy.

For the oryzalin treatment, oryzalin (Sigma-Aldrich, St Louis, MO, USA) was dissolved in dimethylsulfoxide (DMSO) as a 20 mM stock solution, stored at −20 °C, and later diluted to 10 μM final concentration with distilled water. Before observation, discs cut from infiltrated tobacco leaves were incubated in oryzalin solution (or in water for controls). 

Microscopic images were obtained using the Zeiss LSM880 (Carl Zeiss, Oberkochen, Germany) confocal laser scanning microscope with a C-Apochromat 40×/1.2 W Korr FCS M27 objective. Fluorophores were excited with the 488 nm argon laser (YFP, GFP) and 561 nm laser (RFP), respectively, and detected by sensitive 32-chanell Gallium arsenide phosphide (GaAsP) spectral detector. All images were processed and analyzed using the Fiji software. All typical localization patterns that have been regularly observed in two to four independent biological replicates, each involving several leaves, are shown. Particle tracking was performed using the Fiji plugin TrackMate [55] v3.8.0 with the following settings: LoG detector, 2 µm diameter; no median filter; no sub-pixel localization; thresholding values were set to discard false-positive spots while the correctly identified spots were retained; simple LAP tracker; linking max distance = 5 µm; gap closing max distance = 5 µm; gap-closing max frame gap = 2. Data from at least five cells for each treatment with more than 200 trajectories detected were proceeded into a boxplot using the BoxPlotR tool [56]. To quantify protein co-localization, Pearson’s coefficient was calculated from at least five images per construct pair using the Fiji/ImageJ plugin JACoP (Just Another Colocalisation Plugin) with manual threshold correction [57]. For statistic evaluation, an online tool [58] was used to perform one-way ANOVA with post-hoc Tukey HSD (honestly significant difference) test.

Constructs for the yeast two-hybrid assay were prepared by restriction cloning from previously prepared AtFH13 and AtFH14 clones (see above) and introduced into vectors pGADT7 (containing the GAL4 activation domain) and pGBKT7 (containing the GAL4 DNA binding domain). Primers, templates, and destination vectors are summarized in Appendix A. The yeast two-hybrid assay was performed using Matchmaker GAL4 two-hybrid system 3 (Clontech, Mountain View, CA, USA) as described previously [50]. 

## Figures and Tables

**Figure 1 ijms-21-00348-f001:**
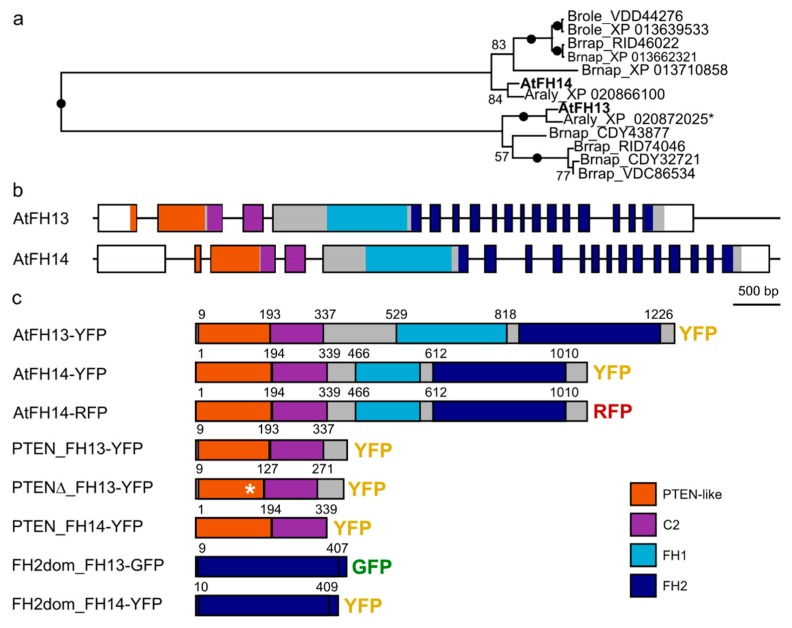
Evolutionary relationships of Arabidopsis Class II formins (AtFH13 and AtFH14), structure of the AtFH13 and AtFH14 genes, and constructs generated in this study. (**a**) Maximum likelihood phylogenetic tree of joined PTEN, C2 and FH2 domains of AtFH13, AtFH14 and their homologs from *Arabidopsis lyrata* (Araly), *Brassica napus* (Brnap), *B. rapa* (Brrap), and *B. oleracea* (Brole). Sequences are identified by their GenBank accession numbers; the asterisk marks a predicted protein sequence modified to include missing conserved exons (see Methods). Numbers denote bootstrap support (out of 100 replicates), branches with 100% support are marked by dots. (**b**) Exon–intron structure of both genes with non-coding (UTR) parts of exons represented by white boxes and coding exons shown either in grey or in color (for known domains). (**c**) Schematic representation of tagged protein constructs used in this study. Asterisk indicates the position of the 21 amino acids deletion in the mutant PTENΔ domain, starting from position 106 of the standard AtFH13 sequence. The numbers indicate amino acid positions related to the N-end. Domain abbreviations are defined in the text.

**Figure 2 ijms-21-00348-f002:**
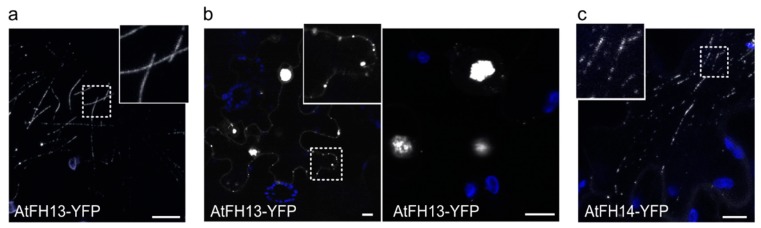
Localization of AtFH13-YFP and AtFH14-YFP in transiently transformed tobacco epidermis. (**a**) AtFH13-YFP localizes to fibrous structures at low expression levels. (**b**) Aggregates of overexpressed AtFH13-YFP. (**c**) Fibrous localization of AtFH14-YFP. All images are maximum intensity projections of confocal Z-stacks. YFP fluorescence is shown in grayscale, chlorophyll autofluorescence in blue. Scale bars are 10 µm, insets show the indicated details magnified 2.5 times compared to the main image. Construct abbreviations are defined in the text and in Figure 1.

**Figure 3 ijms-21-00348-f003:**
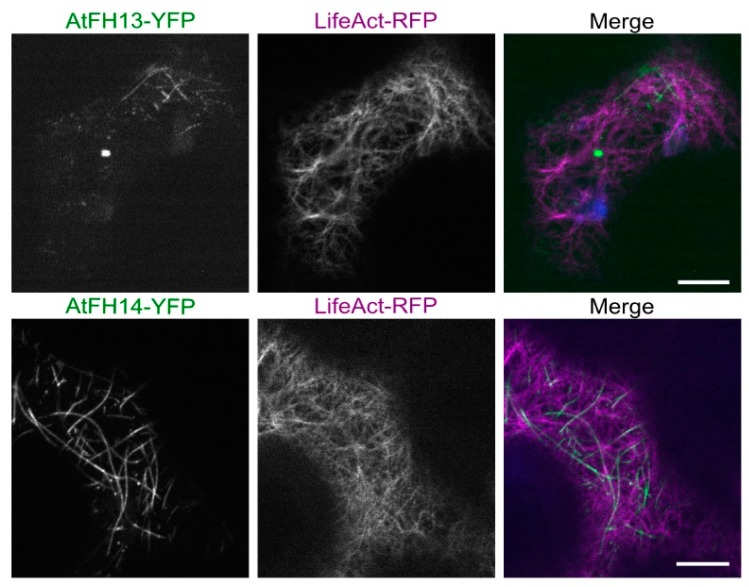
Single optical sections of epidermal cells co-expressing AtFH13-YFP or AtFH14-YFP with the actin marker LifeAct-RFP. Images from single channels are shown in grayscale, merged images display an overlay of three channels represented by green (YFP), magenta (RFP), and blue (chloroplast autofluorescence). Scale bars are 10 µm. Construct abbreviations are defined in the text and in Figure 1.

**Figure 4 ijms-21-00348-f004:**
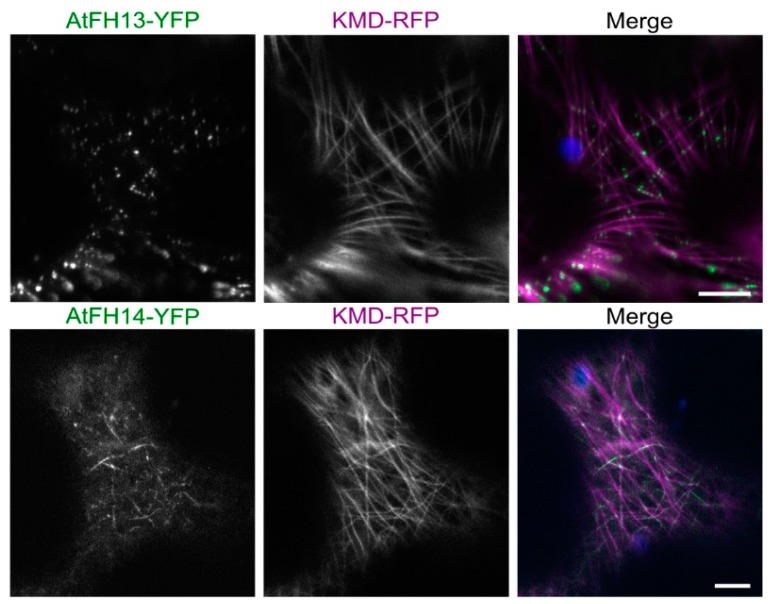
Single optical sections of epidermal cells co-expressing AtFH13-YFP or AtFH14-YFP with the microtubule marker KMD-RFP. Images from single channels are shown in grayscale, merged images display an overlay of three channels represented by green (YFP), magenta (RFP), and blue (chloroplast autofluorescence). Scale bar is 10 µm. Construct abbreviations are defined in the text and in Figure 1.

**Figure 5 ijms-21-00348-f005:**
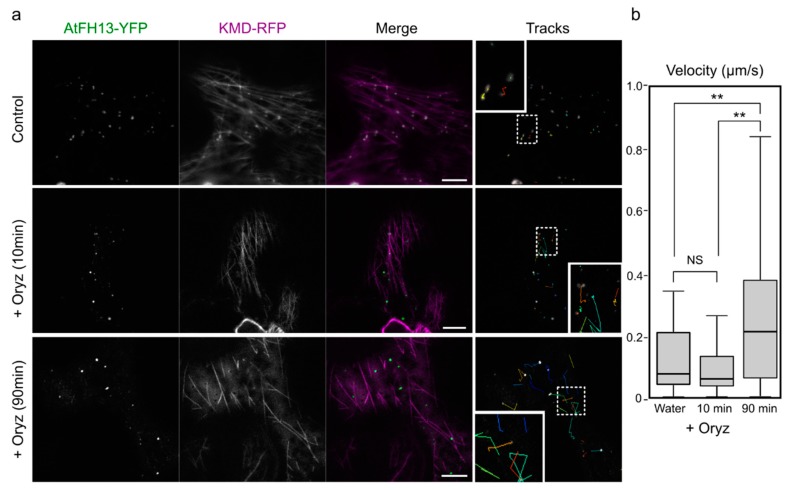
Prolonged oryzalin treatment increases mobility of AtFH13-YFP dots. (**a**) Epidermal cells co-expressing AtFH13-YFP and KMD-RFP mock-treated with water (control) and after 10 μM oryzalin treatment (+Oryz) for 10 min (no microtubule disruption observed) or 90 min (microtubules largely depolymerized). Single channel images are shown in grayscale, merged images display an overlay of the YFP channel in green with RFP in magenta. Particle tracks show trajectories of individual AtFH13-YFP dots over 50 s with insets magnified 2.5 times compared to the main image. (**b**) Box plot of mean velocities of AtFH13-YFP particles without, after 10 and after 90 min of oryzalin treatment. Scale bar is 10 µm. Two asterisks indicate a statistically significant difference (ANOVA, *p* < 0.001), NS: Non-significant. Construct abbreviations are defined in the text and in Figure 1.

**Figure 6 ijms-21-00348-f006:**
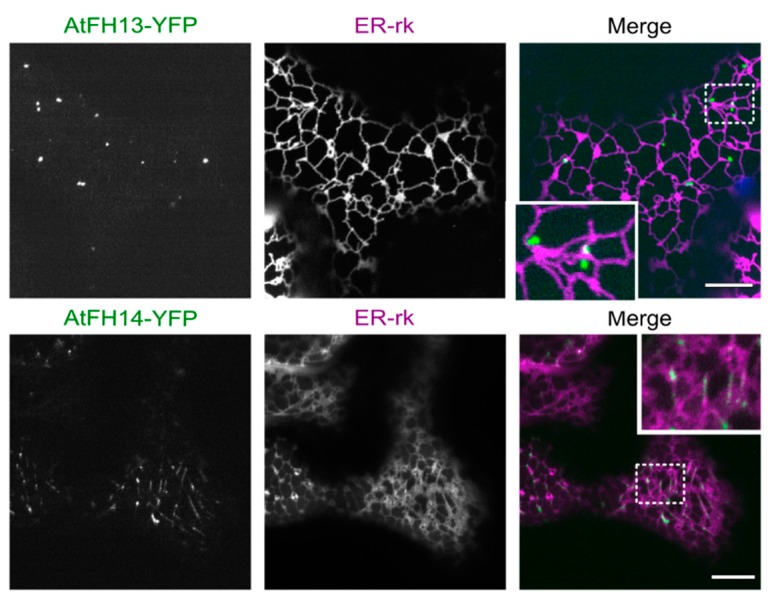
Single optical sections of epidermal cells co-expressing AtFH13-YFP and AtFH14-YFP with the endoplasmic reticulum marker ER-rk. Images from single channels are shown in grayscale, merged images display an overlay of three channels represented by green (YFP), magenta (RFP), and blue (chloroplast autofluorescence). Scale bars are 10 µm, insets are magnified 2.5 times compared to the main image. Construct abbreviations are defined in the text and in Figure 1.

**Figure 7 ijms-21-00348-f007:**
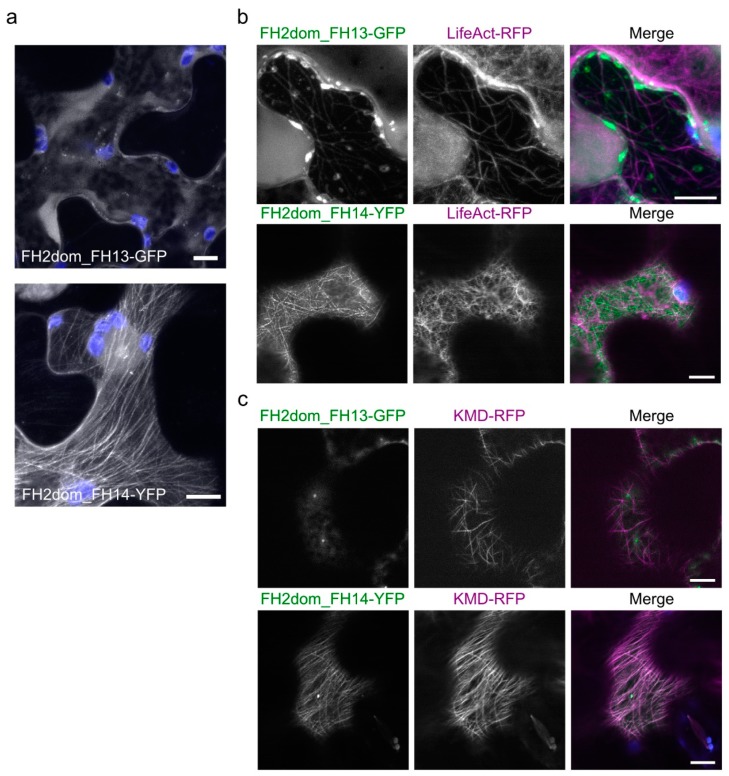
Localization of isolated FH2 domains of AtFH13 and AtFH14 in tobacco epidermal cells. (**a**) Maximal intensity projections of Z-stacks of optical sections from cells expressing FH2dom_AtFH13-GFP and FH2dom_AtFH14-YFP constructs, respectively. Green (GFP) or yellow (YFP) fluorescence shown in grayscale, chlorophyll autofluorescence in blue. (**b**) Single optical sections of cells co-expressing FH2dom_AtFH13-GFP or FH2dom_AtFH14-YFP with LifeAct-RFP. (**c**) Single optical sections of cells co-expressing FH2dom_AtFH13-GFP and FH2dom_AtFH14-YFP with KMD-RFP. In (**b**) and (**c**), images from single channels are shown in grayscale, merged images display an overlay of three channels represented by green (GFP or YFP), magenta (RFP), and blue (chloroplast autofluorescence. Scale bars are 10 µm. Construct abbreviations are defined in the text and in Figure 1.

**Figure 8 ijms-21-00348-f008:**
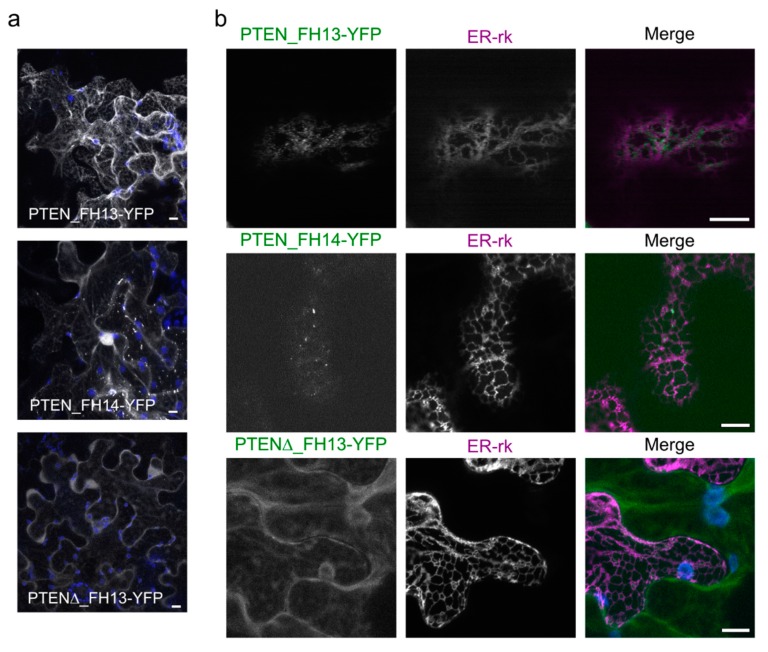
Localization of isolated PTEN domains of AtFH13 and AtFH14 in tobacco epidermal cells. (**a**) Maximal intensity projections of Z-stacks of optical sections from epidermal cells expressing PTEN_AtFH13-YFP, PTEN_AtFH14-YFP, or PTENΔ_AtFH13-YFP. Yellow (YFP) signal shown in grayscale, chlorophyll autofluorescence in blue. (**b**) Single optical sections of cells co-expressing PTEN_AtFH13-YFP, PTEN_AtFH14-YFP, or PTENΔ_AtFH13-YFP with ER-rk. Images from single channels are shown in grayscale, merged images display an overlay of three channels represented by green (YFP), magenta (RFP), and blue (chloroplast autofluorescence). Scale bars are 10 µm. Construct abbreviations are defined in the text and in Figure 1.

**Figure 9 ijms-21-00348-f009:**
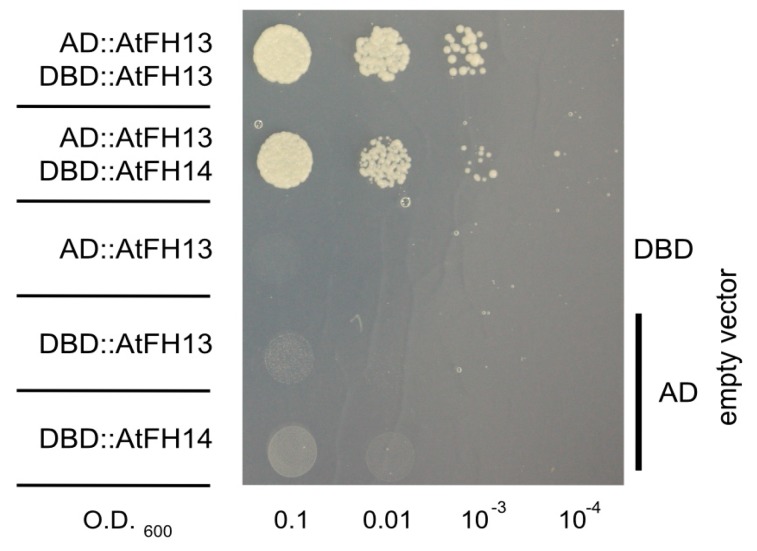
Yeast two-hybrid assays documenting heterodimerization of FH2 domains of AtFH13 and AtFH14. Serial dilution series of yeast transformants harboring the indicated constructs starting with OD_600_ = 0.1 were plated on media without adenine (-Ade), histidine (-His), leucine (-Leu), and tryptophan (-Trp) as indicated. DBD: DNA binding domain; AD: activation domain.

**Figure 10 ijms-21-00348-f010:**
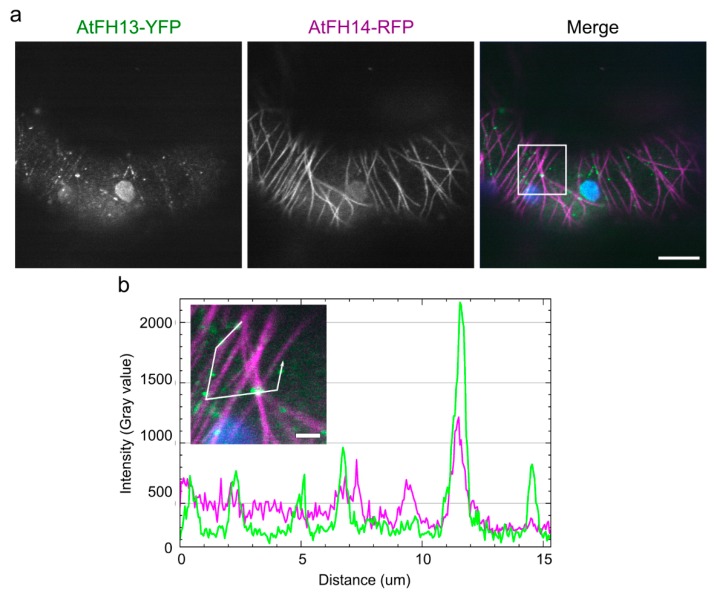
Co-localization of AtFH13 and AtFH14 in transiently transformed tobacco epidermal cells. (**a**) Single optical section of a cell co-expressing AtFH13-YFP and AtFH14-RFP. Images from single channels are shown in grayscale, merged images display an overlay of three channels represented by green (YFP), magenta (RFP), and blue (chloroplast autofluorescence). Note that one of the chloroplasts is probably damaged, exhibiting yellow autofluorescence. Scale bar is 10 µm. (**b**) Measurement of fluorescence intensity profiles of AtFH13-YFP and AtFH14-RFP along the indicated broken line in arrow direction. Scale bar is 1 µm. Construct abbreviations are defined in the text and in Figure 1.

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
