# Peer review of "Arabidopsis Class II Formins AtFH13 and AtFH14 Can Form Heterodimers but Exhibit Distinct Patterns of Cellular Localization"

_ijms, 2020, doi:10.3390/ijms21010348_

Round 1

Reviewer 1 Report

Although this study appears to be well carried out, it seems to be more incremental and possibly limited to plant biology. Overall, the work is mostly structural in nature in terms of identifying where class II formins AtFH13 and 14 localize in plant cells, and of forming apparent heterodimers. 

Aside from this limitation, a few major concerns would be the use of relatively large YFP (or any fluorescent protein), which itself could change various interaction patterns. In this regard, it would be better to use fluorescent tags, like FITC or one of the Alexas, and assess binding and localization in this way. 

Also, the authors do not demonstrate biological relevance for their findings, especially the new report of heterodimerization. 

Therefore, overall the study is quite limited and incremental, and perhaps better suited for a journal on plant biology.    

Author Response

While we agree that our study is mostly descriptive, we do believe that the first report of formin herterodimerization mediated exclusively by the FH2 domain (to our knowledge in any organism!) may be of interest also outside the field of plant biology, even in the absence of documented phenotypic consequences. We have modified the Abstract to emphasize this aspect of our work.

We are aware of possible artifacts associated with the use of fluorescent protein fusions to determine intracellular  localization. However, there are currently no established alternative methods in plant cell biology besides immunostaining of fixed cells using specific antibodies (which we unsuccessfully attempted to generate). Although expression of small tags that can be visualized with the aid of cell permeable ligands is commonly used in mammalian cell biology, in the plant field such an approach has, to our knowledge, been successfully applied only to study extracytoplasmic proteins (with the exception of a single study from 2006 with  no published experimental follow-up, where the  tag is used as a reporter to monitor sites of gene expression rather than for protein localization). We have modified the Discussion to comment on the inherent limitations of fluorescent protein use, as well as the (lack of) practically usable alternatives. 

Reviewer 2 Report

This manuscript outline the subcellular localization of two plant formins and demonstrates for the first time the possibility, via yeast two hybrid of formin heterodimerization.  It is well written and an important addition to the literature.  

Microscopy demonstrates co-localization with single images in the manuscript.  I'd make one request: that there's some sort of quantitation of the co-localization to demonstrate that this is a general phenomenon, not just in the single image shown.  The authors can indicate in the methods how many cells were images, that the image is a representative image, and discuss what it looked like generally, as well as some quantitative co-localization.

Author Response

Thank you for these  suggestions!

Information on the number of biological replicates and criteria for typical image selection have been added to the Methods.

We also added  some quantitative colocalization data as a new Supplementary figure. Although the observed differences are not statistically significant (possibly because of a relatively low signal to noise ratioin our images, which means that the method used is operating at its limits), they do indicate trends consistent with our qualitative descriptions.

Reviewer 3 Report

The Manuscript by Kollárová et al. reports on formins heterodimerization mediated by FH2 domains. Formins are essential players in the regulation of the cytoskeleton and cell polarity in all eukaryotes. The fundamental results of the research reported by authors include the demonstration of FH2-mediated heterodimerization of AtFH13 and AtFH14 and the comparison of their intracellular distribution patterns. Taking into account that the fluorescence imaging in plants is a challenging task, the Referee is satisfied with the overall quality of primary data and the analysis presented in the Manuscript. 

The paper is concise, mature, and well written. The Discussion is excellent and uses accurate and soft language in concluding. The Authors are cautious and understand the limitations of the methods used in the study.  

Some figures could be improved. 

The Referee would like to share the following comments on the Manuscript:

Please check the consistency of using abbreviations in the text. For instance, “FH13” and “FH14” are used in some places, including the Abstract. Please use AtFH13 and AtFH14 instead Source for DEST vectors should be cited explicitly (is it 10.1111/j.1365-313X.2010.04322.x ?) to avoid confusion over the exact sequence of genes hiding behind GFP, YFP, and RFP abbreviations (i.e., LifeAct-mRFP: is it mRFP or mRFP1 or mRFPRuby like in Nat.meth paper about LifeAct?). This is especially interesting in conjunction with aggregates observed in Figure 2b. Improper naming or incomplete reference to the exact fluorescent proteins used in plant research is a custom that hinders tracking the performance of various fluorescent proteins in plant hosts by resources like  https://www.fpbase.org/ The supplementary videos are beneficial. Consider providing the files separately and not in a single zip. Please also check the compatibility of codecs used in .avi - the ones in S3 and S4 differ from S1 and S2, and require some tricks in order to get them played on some platforms. The Referee suspects the difference starts with the ‘compression’ setting (JPEG/none) in the ‘save as AVI’ command of ImageJ.  The Referee has followed the chain of ‘as described previously’ links for transient transformation conditions starting from ref [50]. The Referee sincerely believes that the time has come to articulate the conditions again explicitly, including strains and buffer composition. Figure 2 is barely readable in its current form. Consider adding magnified panel or insets. Magnified insets would also be beneficial for some other figures with minuscule details.  Figure 10a-Merge. The coexistence of blue and cyan chloroplasts is strange, assuming linear ‘blue’ colormap for the autofluorescence. Consider adding color bar if the lookup table for the pseudocolor is not linear Figure 1b,c. The shadows (or blur?) in “YFP,” “RFP,” and the asterisk symbol decrease the readability of the figure.  Figure 1b,c. The color palette is not a colorblind-friendly one.

Author Response

Thank you for a very careful review that indeed identified some flaws and inaccuracies which we hopefully have now removed , as far as it was possible, i. e. except point (3), as follows.

(1) (In)consistency of using abbreviations:

Thanks for noticing the mis-named proteins - the errors have been corrected.

(2) Vectors and fluorescent proteins description:

Thanks for alerting us to the incomplete documentation that, moreover, led us towards discovering an  error in the description of one of the fluorophores used in our study, The error has now been corrected both in the text and in the figures, and additional information on the constructs used has been included.

(3) Supplementary files in a single zip archive:

We would have also preferred separate supplementary files. Unfortunately, this is not permitted by the journal.

(4) Compatibility of video codecs:

Because of file size limits, some of the videos required additional compression before upload. However, in the absence of more detailed information about the reviewer´system and the errors encountered, we were unable to reproduce the issues reported by the reviewer, even on multiple computers running different OS. We have thus at least added information on compression methods for individual videos in the supplementary data legends.

(5) Description of the transient transformation method:

We have now included a detailed description of our technique (and cleaned away some inaccuracies that arose in the long citation chain).

(6) Barely readable details in some figures:

Insets with magnified details have been added to Figures 2, 5 and 6.

(7) Chloroplast fluorescence in the YFP channel in Fig. 10:

The color map in the figure is linear but we believe that we happened to capture a damaged chloroplast exhibiting yellow autofluorescence (absent in other plastids within the same image). A comment has been added to the figure legend.

(8) Figure 1 color and graphics:

Figure 1 has been modified according to the reviewers´ suggestions (using a color palette selected with the aid of the Coblis software (see https://www.color-blindness.com/coblis-color-blindness-simulator/).

We also performed an additional round of spelling check.

Round 2

Reviewer 1 Report

The authors have made some improvements to their manuscript as outlined in the initial review. However, my initial concerns remain in that the study is only an incremental advance in the field and it lacks biological relevance. It is also unclear whether the readers of the Journal would find the work of great interest, thus limiting the number of citations.  

Reviewer 2 Report

The authors performed the analysis I requested.  I believe this paper should be accepted for publication.